# Chronic Heat Stress Affects Bile Acid Profile and Gut Microbiota in Broilers

**DOI:** 10.3390/ijms241210238

**Published:** 2023-06-16

**Authors:** Yuting Zhang, Huimin Chen, Wei Cong, Ke Zhang, Yimin Jia, Lei Wu

**Affiliations:** Key Laboratory of Animal Physiology & Biochemistry, College of Veterinary Medicine, Nanjing Agricultural University, Nanjing 210095, China; zhangyt001210@163.com (Y.Z.); 2021107035@stu.njau.edu.cn (H.C.); 2019107035@njau.edu.cn (W.C.); 2022107011@stu.njau.edu.cn (K.Z.); jymrobin@hotmail.com (Y.J.)

**Keywords:** heat stress, bile acid, gut microbiota, enterohepatic circulation, broiler

## Abstract

Heat stress (HS) can inhibit the growth performance of broilers and cause substantial economic losses. Alterations in bile acid (BA) pools have been reported to be correlated with chronic HS, yet the specific mechanism and whether it is related to gut microbiota remains unclear. In this study, 40 Rugao Yellow chickens were randomly selected and distributed into two groups (20 broilers in each group) when reaching 56-day age: a chronic heat stress group (HS, 36 ± 1 °C for 8 h per day in the first 7 days and 36 ± 1 °C for 24 h in the last 7 days) and a control group (CN, 24 ± 1 °C for 24 h within 14 days). Compared with the CN group, total BAs’ serum content decreased, while cholic acid (CA), chenodeoxycholic acid (CDCA), and taurolithocholic acid (TLCA) increased significantly in HS broilers. Moreover, 12α-hydroxylase (CYP8B1) and bile salt export protein (BSEP) were upregulated in the liver, and the expression of fibroblast growth factor 19 (FGF19) decreased in the ileum of HS broilers. There were also significant changes in gut microbial composition, and the enrichment of *Peptoniphilus* was positively correlated with the increased serum level of TLCA. These results indicate that chronic HS disrupts the homeostasis of BA metabolism in broilers, which is associated with alterations in gut microbiota.

## 1. Introduction

As the global temperature rises continually and the breeding intensity increases, HS has become a common threat to poultry, leading to enormous economic losses to poultry husbandry. HS has been shown to induce severe adverse effects on growth performance, feed intake, egg production/quality, and increased mortality in poultry [1]. Studies have also shown that HS disturbs lipid metabolism, leading to extra fat deposition in the abdomen and muscle, further lowering meat quality in the poultry industry [2,3]. Thus, clarifying the mechanisms of HS-induced lipid metabolism disruption is essential to ensure the sustainable development of poultry husbandry and decrease the annual economic loss caused by HS. As an emulsifier of dietary lipids, bile acids (BAs) were speculated to be associated with lipid metabolism disorder during chronic HS [4,5]. BAs are primarily synthesized in the liver and further metabolized by the gut microbiota. Therefore, enterohepatic circulation plays an irreplaceable role in BA metabolism, and dysbiosis in gut microbiota during HS has also been observed in broilers [6,7], indicating that alterations in gut microbiota might cause BA metabolism disorder during HS. However, at present, it is not entirely known what the specific relationship between BA composition and gut microbiota during HS is, and the mechanism remains unclarified.

Primary BAs are initially synthesized in the liver from cholesterol in two ways. The primary pathway is the main BA synthesis pathway and is activated by cholesterol 7α-hydroxylase (CYP7A1) on the smooth endoplasmic ream of hepatocytes. In this pathway, 12α-hydroxylase (CYP8B1) is the essential enzyme for synthesizing cholic acid (CA) [8]. Otherwise, chenodeoxycholic acid (CDCA) is produced without CYP8B1 [9]. The alternative pathway is initiated by sterol 27-hydroxylase (CYP27A1) on the mitochondria and catalyzed by oxysterol 7-α-hydroxylase (CYP7B1) to synthesize CDCA [10]. CA and CDCA in the liver combine with glycine or taurine to form bile salts after synthesis. Bile salts are then transported into the gallbladder via the bile duct by bile salt export protein (BSEP) and multidrug-resistance-related protein 2 (MRP2) or transported into systemic circulation by organic solute transporter α and β dimer (OSTα/OSTβ). Bile salts are released into the intestine after food intake [11,12]. Gut microbiota deconjugate the glycine or taurine-conjugated BAs to form free BAs, which are further converted to secondary BAs by 7α-dehydroxylase in the gut [13]. About 95% of bile salts are actively reabsorbed via apical sodium-dependent transporter (ASBT) at the end of the ileum and then excreted into the portal vein by OSTα and OSTβ in the basal layer of the ileum [13,14]. Secondary BAs are either passively absorbed into the portal vein or excreted in feces. Reabsorbed primary and secondary BAs circulate through the portal vein to the hepatic sinusoid. They are transported to hepatocytes by sodium taurocholate transporting polypeptide (NTCP) and organic anion transporters (OATPs), respectively, to complete an enterohepatic circulation [15]. Gut microbiota not only affects the composition of the BA pool but also adjusts the expression of the farnesoid X receptor (FXR), a BA-activated receptor [16]. Furthermore, the FXR/small heterodimer partner (SHP) and the FXR/fibroblast growth factor 15/19 (FGF15/19) signaling pathways are closely related to the negative feedback regulation of BA metabolism [17].

Many studies have noticed that specific types of BAs could decrease after chronic HS. For instance, Wei Fang et al. observed that exposing large white pigs to 33 °C for 21 days significantly inhibited hepatic BA synthesis, conjugation, and uptake transport, thereby reducing serum taurine-conjugated BA (TCBA) levels [18]. Until now, alterations in beneficial gut microbiota after chronic HS were noticed in several studies, such as decreased levels of *Coprococcus* and *Aeriscardovia* [19]. Moreover, the dietary supplementation of bile salts at 4.5 mg/kg promotes fat catabolism and improves fat deposition in broilers [20]. The BA supplementation was also found to potentially improve the antioxidant capacity of broilers exposed to high temperatures by regulating the structure of the gut microbiota, which might be a novel nutritional strategy against HS [21]. However, previous work rarely focused on the correlation among chronic HS, gut microbiota, and BA metabolism, especially on broilers.

Above all, alterations in gut microbiota structure and BA profiles are suspected to be reasons for disrupted lipid deposition and reduced growth performance during chronic HS. In this study, we aimed to observe these alterations and focus on the relationship between BAs and gut microbiota in broilers. Therefore, our findings may provide a reference for elucidating the mechanism to guide adding BAs to diets as a nutritional strategy for alleviating chronic HS by altering the structure of the gut microbiota.

## 2. Results

### 2.1. Chronic HS Inhibits Growth Performance and Leads to Hepatic Lipid Deposition in Broilers

Our study on growth performance showed that compared to the CN group, broilers exposed to chronic HS had a significant reduction in body weight on D70 and average daily feed intake (*p* < 0.01, Figure 1A). The neutrophil-to-lymphocyte (H/L) result showed that chronic HS could effectively increase the ratio of H/L (*p* < 0.01) and decrease the erythrocyte hematocrit (*p* < 0.05) in the blood (Figure 1B). However, chronic HS did not affect the body temperature and plasma total antioxidant capacity (Figure 1A,B). Moreover, compared with the CN group, the plasma concentration of AST and LDH significantly increased after chronic HS (*p* < 0.05). On the contrary, chronic HS significantly decreased the plasma concentration of ALP and GLU (*p* < 0.05, Table 1).

Small lipid droplets can be observed in the representative images of chronic HS liver H&E sections. The loose and disorganized arrangement of liver cells also indicated that chronic HS could cause damage to the liver (Figure 1C). In addition, higher hepatic cholesterol and triglyceride levels could be detected (*p* < 0.05) in the HS group.

### 2.2. Chronic HS Led to Intestinal Inflammation and Disrupted Gut Microbiota Composition

HE images showed inflammatory cell infiltration and bleeding in intestinal tissue after chronic HS treatment (Figure 2A). In addition, the principal coordinates analysis (PCoA) of the cecal microbiota 16S rRNA sequencing results showed an apparent separation of HS from CN (Figure 2B). Furthermore, the α-diversity index (Pielou_e) significantly differed between the CN and HS groups (Figure 2C). Finally, heatmap analysis showed the different microbiota compositions between CN and HS groups at the genus level (Figure 2D). In both the CN and HS groups, the abundance of *Lactobacillus* and *Streptococcus* accounted for most of the cecal gut microbiota. However, the abundance of *Subdoligranulum* (*p* < 0.01), *uncultured_Hyphomonadaceae*, *Blautia*, *Acinetobacter*, *Prauserella*, and *Methylobacterium* (*p* < 0.05) was all significantly lower in the HS group, compared with the CN group (Figure 2E).

According to LEfSe analysis, chronic HS significantly increased the abundance of *Firmicutes* at the phylum level. However, at the genus level, the abundance of *Subdoligranulum*, *Lactococcus*, *Ruminococcus_torques_group*, and *Leucobacter* of the HS group reduced significantly in the cecum (Table 2).

### 2.3. Chronic HS-Disrupted BAs Metabolism in the Liver and Serum

Compared to the CN group, serum TBA significantly decreased in the HS group (*p* < 0.05, Figure 3A). Thus, the serum was subjected to targeted metabolomic analysis of BAs. Moreover, the PCA results showed that the BA profile in the HS group was well separated from the CN group (Figure 3B). Differently, the serum levels of specific primary BAs (CA, *p* = 0.037; CDCA, *p* = 0.006) and secondary BAs (TLCA, *p* = 0.047) were significantly upregulated in the HS group (Figure 3C,D).

Primary BAs are mainly synthesized in the liver, condensed in the gallbladder, and finally excreted into the small intestine. Thus, the expression of genes related to BA metabolism in the liver and small intestine was tested. The hepatic expression of CYP8B1 (*p* < 0.01) and NTCP (*p* < 0.05) was significantly increased in the HS group compared to the CN group (Figure 4A). However, there was no significant effect on others regarding genes.

It was shown that BA content in serum is also influenced by intestinal metabolism, and the ileum is the leading site of BA reabsorption. Therefore, we also detected the expression of the related genes in the ileum. However, chronic HS did not significantly affect the critical genes of BA reabsorption and enterohepatic circulation apart from FGF19 (*p* < 0.05, Figure 4B).

### 2.4. Alterations in Gut Microbiota Composition Are Associated with BA Metabolism

Studies show that the gut microbiota plays a critical role in BA metabolism. Thus, the correlation between the gut microbiota and serum BAs was conducted using R. At the genus level, the changes in cecum microbial concentration were significantly correlated with serum BA after chronic HS. The abundance of *Gallicola* was positively correlated with the CDCA (*p* < 0.01) and CA (*p* < 0.05) levels in serum. The abundance of *Peptoniphilus* has positively correlated with CDCA (*p* < 0.01), HDCA, CA, TLCA, and THDCA/TUDCA (*p* < 0.05) levels in serum (Figure 5).

## 3. Discussion

The treatment of 36 ± 1 °C for 8 h per day in the first 7 days and 36 ± 1 °C for 24 h in the next 7 days could induce chronic HS with reduced body weight and average daily feed intake, which is consistent with previous studies [4,21]. As a sensitive index of stress in broilers, the significant increase in H/L indicated that broilers were stressed by chronic heat treatment [22,23]. Furthermore, the significant reduction in HCT was also a sign of HS in broilers [24]. When broilers are subjected to chronic stress, the liver cannot properly utilize cholesterol to produce BAs, resulting in increased cholesterol and decreased BAs in the liver [25]. The present study’s representative images of liver H&E staining and biochemical results illustrated that chronic HS led to extra lipid deposition in the liver. In addition, the serum-targeted BA metabolomics showed that the content of CA, CDCA, and TLCA increased significantly while the content of total BAs decreased considerably. As a result, we indicate that chronic HS can destroy the metabolic homeostasis of BAs.

Because BAs are synthesized in the liver and mainly reabsorbed in the ileum [9,10], we measured mRNA expressions of BA-related genes in these two tissues. Contrary to the down-regulated trends on large white pigs [5,18], the expression of CYP8B1, essential for synthesizing CA through the classical pathway, was considerably upregulated after chronic HS in the present study. On the other hand, FGF19, which functions as a negative feedback gene in enterohepatic circulation, was downregulated to promote hepatic primary BA synthesis because of the decrease in serum TBA [26]. As a result, the decreased expression of FGF19 in the ileum might be a reason for the upregulation of CYP8B1 in the liver. Another reason we assumed for the decrease in FGF19 expression could be that disrupted BA metabolism led to diarrhea, and a large amount of BA salts were excreted with the feces [27]. However, the amount of BA excreted in the feces was not measured in the current study. Furthermore, the expression of BA-binding transporter BSEP was upregulated in the HS group, indicating that BAs were excreted more from the liver to the gallbladder [8]. Thus, BAs may tend to be excreted into the gallbladder after chronic HS, resulting in a decrease in TBA in serum.

Given the integral role of gut microbiota in BA metabolism, the cecal microbiota 16S rRNA sequence analysis was also applied to explain the alteration of the BA profile in serum. On the genus level, it has been reported that *Bacteroides*, *Clostridium*, *Listeria*, *Lactobacillus*, and *Bifidobacterium* are essential for carrying out the deconjugation of primary BAs; *Clostridium*, *Eubacterium*, and *Ruminococcus* are significant players in the oxidation and epimerization of the 3-, 7-, and 12-hydroxy of primary BAs; *Clostridium* and *Eubacterium* mainly participate in the 7-dehydroxylation [28]. It was noticed in a previous study about broilers that chronic cyclic HS increased the abundance of *Tyzzerella* and decreased the abundance of *Bacteroides*, *Parabacteroides*, and *Romboutsia* in the cecum of broilers [29]. In the present study, alterations in other genera were observed, such as the enrichment of *Gallicola* and *Peptoniphilus* and the reduction in *Subdoligranulum*, *Ruminococcus_torques_group*, and *Ruminococcaceae_UCG-014*.

Although it is now established that the microbiome residing in the gastrointestinal tract plays a significant role in promoting the host’s overall health, a considerable number of microbiomes constituting the community remain uncultured or unclassified. Thus, we conducted a correlation analysis between serum BA and the gut microbiome. The results showed that the increase in TLCA content in serum was positively related to the enrichment of *Peptoniphilus* in the cecum. CDCA is modified by the gut microbiota into LCA and becomes TLCA when combined with taurine [30]. Thus, it is speculated that *Peptoniphilus* plays a particular role in the formation of secondary BAs from primary BAs. Moreover, the abundance of *Gallicola* was positively correlated with the CDCA and CA levels in serum. However, due to the difficulty of culturing, seldom studies were found to define the connection between *Gallicola* and primary BAs synthesis. Future studies are needed to study the role of *Gallicola* in BA metabolism. In conclusion, the results of the current study indicate that chronic HS-induced gut microbiota dysbiosis could account for the disruption of BA metabolic homeostasis in broilers.

In this study, chronic HS led to disorders of BA metabolism, inducing lipid metabolism disorders and decreased growth performance in broilers—the alterations in gut microbiota after stress might be a reason for the changes in the BA pool. However, the current study focused on the serum-targeted metabolomics of BA, and the excretion of BAs into feces was unfortunately ignored. Moreover, the BA pool kept circulating dynamically while we only examined the changes in BAs and gut microbiota simultaneously. Therefore, subsequent studies are needed to focus on the dynamic changes to understand BA metabolism mechanisms better and determine the specific functions of gut microbiota in BA metabolism.

## 4. Materials and Methods

### 4.1. Animals and Experimental Design

One hundred Rugao Yellow chicken eggs were purchased from the Jiangsu Institute of Poultry Sciences (Yangzhou, China). Unfertilized and dead eggs were removed at E18 and then incubated until hatching under 37 °C and 65% humidity. After hatching, one-day-old broilers were individually weighed, wing labeled, and raised following the standard recommended by the breeder. The temperature was 35 ± 1 °C during the first week and then decreased by 3 °C every week to 24 ± 1 °C. The relative humidity was kept at 40–60%, and the management procedures (feeding, ventilation, lighting, etc.) adhered to the Feeding Management Regulations of Rugao Chickens. At 56 d of age, 40 chickens were selected and divided into two groups (20 in each group), which were subjected to either control (CN) or chronic heat stress (HS) for 14 d. For the HS group, all chickens were subjected to 36 ± 1 °C for 8 h per day in the first 7 days and 36 ± 1 °C for 24 h in the last 7 days. At the same time, broilers in the CN group were raised under the normal temperature of 24 ± 1 °C (Appendix A).

After 14 days of chronic HS treatment, broilers fasted for 10 h before sample collection, during which, they drank freely. All the broilers were weighed and killed by rapid decapitation, which complies with the American Veterinary Medical Association (AVMA) Guidelines for the Euthanasia of Animals: 2013 Edition. Blood samples were collected into the anticoagulation tube with sodium citrate and centrifugal to separate plasma samples stored at −20 °C. Livers (without the gallbladder), cecal contents, and ileal mucosa were collected and frozen in liquid nitrogen. At the end of the necropsy, all the samples were saved under −80 °C for further analysis.

### 4.2. Determination of Neutrophil to Lymphocyte (H/L) and Erythrocyte Hematocrit (HCT)

The change in the H/L ratio reflects the stress level of broiler chickens and can be used as an indicator to detect the stress level of broiler chickens. First, blood smears were collected from the wing vein and stained with Wright’s stain (G1040, Solabio, Beijing, China) and Giemsa stain (G1010, Solabio). Next, 100 white blood cells were randomly selected, including lymphocytes and eosinophilic granulocytes, in the field of view under the microscope, and the ratio of eosinophilic granulocytes to lymphocytes was calculated.

### 4.3. Measurement of Hepatic Cholesterol (CHOL), Triglyceride (TG), and Serum Total Bile Acids (TBA)

Hepatic CHOL and TG were determined using commercial assay kits (E1015 and E1013, Applygen Technologies Inc., Beijing, China) following the manufacturer’s instructions. The commercial assay kit (E003-2) for testing TBA was provided by Nanjing Jian Cheng Bioengineering Institute, Nanjing, China. Serum TBA was determined by the automatic biochemical analyzer (HITACHI-7020, Hitachi, Tokyo, Japan) using a commercial kit (H101T), following the manufacturer’s instructions.

### 4.4. Determination of Plasma Biochemical Indicators Total Antioxidant Capacity (T-AOC)

Plasma levels of alanine aminotransferase (ALT), aspartate aminotransferase (AST), alkaline phosphatase (ALP), glucose (GLU), total cholesterol (T-CHO), TG, and lactate dehydrogenase (LDH) were determined by an automatic biochemical analyzer (HITACHI-7020, Hitachi, Japan) using commercial kits (H001, H002, H003, H108, H202, H201, H008), according to the manufacturer’s instructions. In addition, t-AOC in plasma was accessed using the commercial kit (A015-2, Nanjing Jian Cheng Bioengineering Institute, Nanjing, China) according to the manufacturer’s instructions.

### 4.5. Sections and Hematoxylin & Eosin (H&E) Staining of Liver and Duodenum

During sampling, the livers and duodenums were cut into pieces at the same location and fixed in 4% paraformaldehyde. The tissue fixation solution was changed 24 h later. Three samples were randomly selected from each group, and the tissues were embedded and sliced using the classical histological experimental method in the laboratory and then stained by H&E as previously described [31].

### 4.6. Serum Targeted Metabolomics of BA

The BA profiles in serum were quantified as previously validated LC-MS procedures [32,33]. Appropriate serum samples were accurately removed into a 2 mL EP tube, and 600 μL methanol (−20 °C) was added for 60 s and then centrifuged at 12,000 rpm for 10 min at 4 °C. Briefly, 400 μL of the supernatant was removed and concentrated to dry, and 200 μL of the sample was redissolved in 30% methanol and vortexed for 30 s [34]. Then, 90 μL of the supernatant was taken into the detection flask for detection by ACQUITY UPLC^®^ BEH C18 column (2.1 × 100 mm, 1.7 μm, Waters, Milford, MA, USA). The mobile phase flow rate was 0.25 mL/min, and the injection volume was 5 μL. MS detection of BAs was conducted in negative mode and under multiple reaction monitoring (MRM) scan mode. The above BA metabolomic analysis was carried out by BioNovoGene, Inc. (Suzhou, China).

### 4.7. Gut Microbiota Analysis

The bacterial DNA from the cecal contents (7 samples/group) was quantified by Nanodrop, and the quality of DNA samples was detected by 1.2% agarose gel. The V3 and V4 hypervariable microbial 16S rDNA regions the DNA samples in the cecum contents, and the primer sequence was by PCR using the following primers: 38F (5′-ACTCCTACGGGAGGCAGCAG-3′); 806R (5′-GGACTACHVGGGTWTCTAAT-3′). The Quant-iT PicoGreen dsDNA Assay Kit and Microplate reader (BioTek, FLx800, Winooski, VT, USA) were used for fluorescence quantification. The PCR products were purified using magnetic beads 0.8 times the volume (VAHTSTM DNA Clean Beads, Vazyme, Nanjing, China). The PCR products were then quantified using the reagent (Quant-iT PicoGreen dsDNA Assay Kit) on the microplate reader (BioTek, FLx800). The library was prepared using TruSeq Nano DNA LT Library Prep Kit (Illumina, San Diego, CA, USA). The MiSeq/NovaSeq (Illumina, USA) platform was used in paired-end DNA sequences, and the raw data were processed with QIIME2 (2019.4) software. The above 16S rRNA sequencing analysis was performed by BioNovoGene, Inc. (Suzhou, China).

### 4.8. RNA Isolation and Real-Time PCR

Thirty microgram liver samples and 20 mg ileal mucosa samples were used to isolate total RNA using 1 mL TRIzol Reagent (TSP401, Tsingke, Beijing, China). Total RNA was reverse-transcribed to cDNA and subjected to real-time PCR with a QuantStudio 6 Flex Real-Time PCR System (Applied Biosystems, Foster City, CA, USA). The primers (Appendix A) were synthesized by Tsingke Biotechnology Co., Ltd. (Nanjing, China). In addition, 18s RNA was used as an internal control. All samples were run in duplicates. Data were analyzed using the method of 2^−∆∆Ct^.

### 4.9. Statistical Analysis

All values were statistically processed and plotted by GraphPad Prism 8.0 (San Diego, CA, USA). Student’s *t*-test was used to compare the treatments. A *p*-value ≤ 0.05 was considered significant. Correlations between serum BAs and gut microbiota abundance were calculated using Pearson’s and Mantel’s correlation analysis. Values were presented as mean ± SEM.

## Figures and Tables

**Figure 1 ijms-24-10238-f001:**
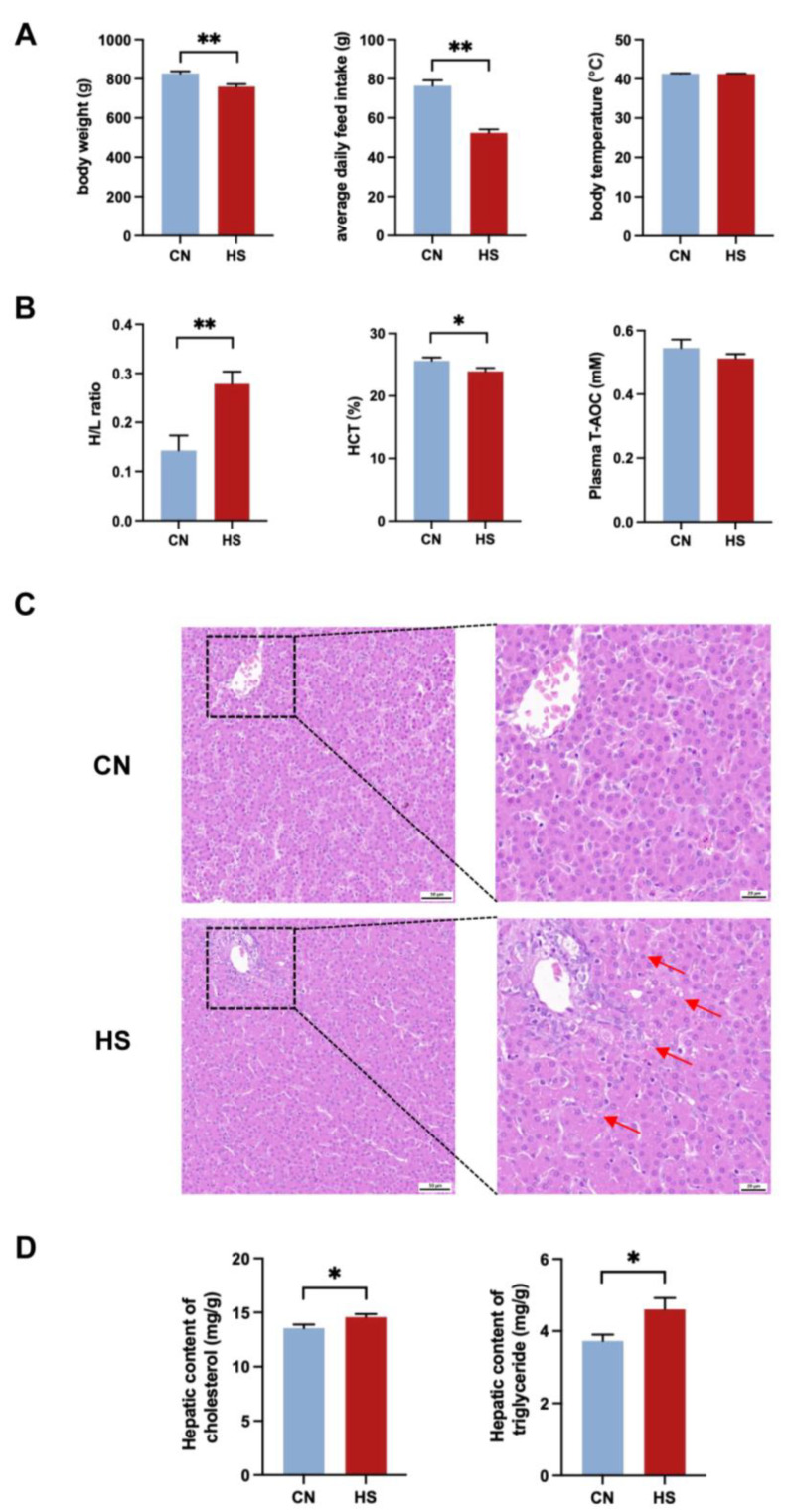
Effect of chronic HS on growth performance and plasma stress-related indicators in broilers. (**A**) Body weight, average daily feed intake, and body temperature; (**B**) H/L, HCT, and plasma total antioxidant capacity (T-AOC); (**C**) representative images of liver H&E sections; (**D**) liver cholesterol, and triglyceride content. Red arrows in (**C**) point to the small fat droplets. Values are mean ± SEM, n = 10, all male. All data were analyzed using Student’s *t*-test. * *p* < 0.05, ** *p* < 0.01.

**Figure 2 ijms-24-10238-f002:**
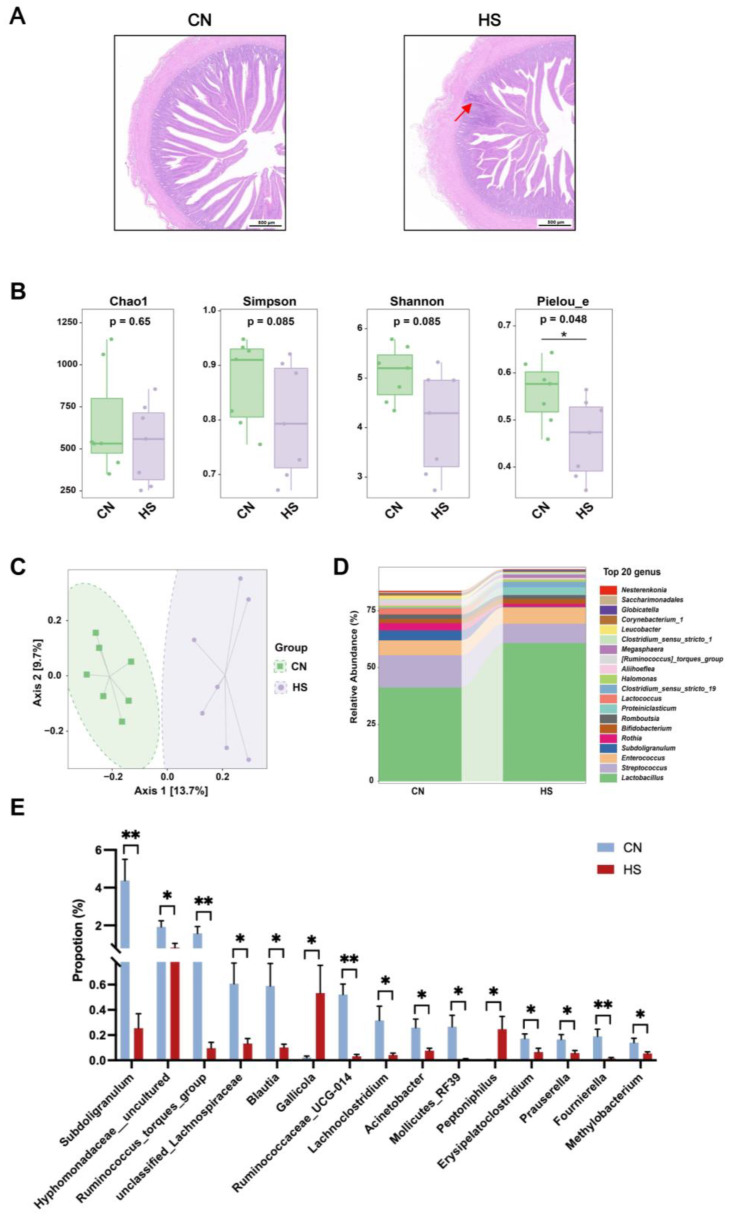
Shifts in cecal microbiota structure after chronic HS. (**A**) Representative images of duodenum H&E section. The red arrow indicated inflammatory cell infiltration and bleeding in intestinal tissue after chronic HS treatment; (**B**) the gut microbiota composition at α-diversity levels, including Chao1, Simpson, Shannon, and Pielou’s evenness; (**C**) the gut microbiota composition at β-diversity levels, PCoA; (**D**) the relative abundance of top 20 genera; (**E**) genera with a significant difference. Values are mean ± SEM, n = 7, all male. Values in (**C**) were analyzed using the Kruskal–Wallis rank sum test and Dunn’s test at α-diversity levels. Values in (**E**) were analyzed using Student’s *t*-test. * *p* < 0.05, ** *p* < 0.01.

**Figure 3 ijms-24-10238-f003:**
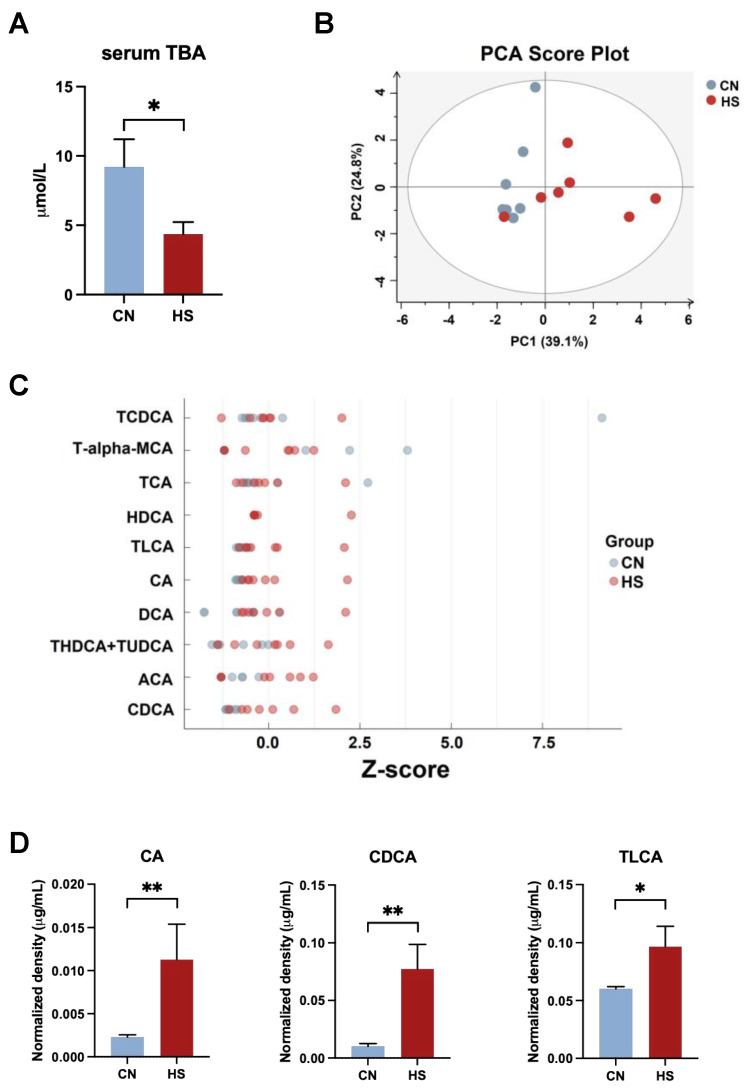
Serum-targeted metabolomics on the alteration of BAs composition in broilers. (**A**) Serum TBA; (**B**) PCA score plots; (**C**) Z-score of BA; (**D**) BA with a significant difference. Values are mean ± SEM, n = 7, all male. All data were analyzed using Student’s *t*-test. * *p* < 0.05, ** *p* < 0.01.

**Figure 4 ijms-24-10238-f004:**
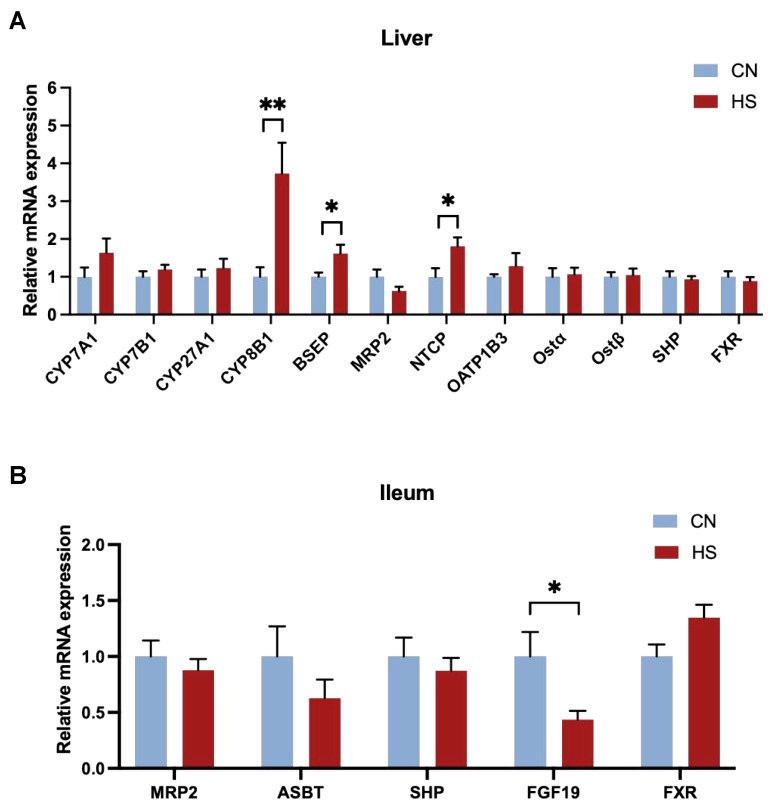
Relative mRNA expression of BA metabolism in intestine and liver. (**A**) Relative mRNA expression of BA metabolism in the liver; (**B**) relative mRNA expression of BA metabolism in the ileum. Values are mean ± SEM, n = 8, all male. All data were analyzed using Student’s *t*-test. * *p* < 0.05, ** *p* < 0.01.

**Figure 5 ijms-24-10238-f005:**
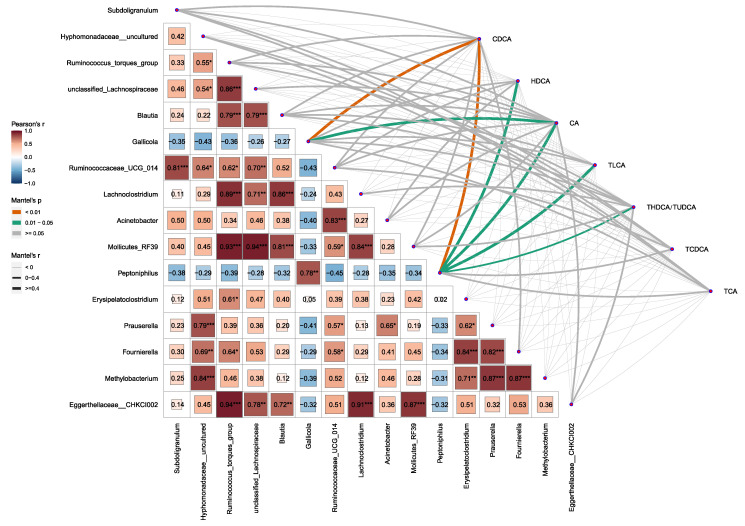
Correlations between serum BAs and abundance of gut microbiota. Values are mean ± SEM, n = 6, all male. Values of correlation analysis between BAs were analyzed using Pearson’s correlation analysis. Values of correlation analysis between BAs and gut microbiota were analyzed using Mantel’s correlation analysis. BAs-related data were analyzed using Student’s *t*-test. * *p* < 0.05, ** *p* < 0.01, *** *p* < 0.001.

**Table 1 ijms-24-10238-t001:** Effects of chronic HS on plasma biochemical indexes in broilers.

	Control	HS	*p*-Value
ALT (U/L)	8.10 ± 1.37	9.20 ± 1.93	0.159
AST (U/L)	209.20 ± 23.63	245.38 ± 44.95 *	0.043
AST/ALT	26.57 ± 5.50	23.79 ± 10.10	0.459
ALP (U/L)	936.90 ± 442.13	581.67 ± 206.72 *	0.042
GLU (mmol/L)	11.44 ± 0.88	10.36 ± 0.89 *	0.020
T-CHO (mmol/L)	3.30 ± 0.82	3.55 ± 0.68	0.526
TG (mmol/L)	0.66 ± 0.13	0.60 ± 0.04	0.170
LDH (U/L)	391.25 ± 42.09	507.11 ± 112.97 *	0.016

Values are mean ± SEM, n = 10, all male. All data were analyzed using Student’s *t*-test. * *p* < 0.05.

**Table 2 ijms-24-10238-t002:** Top 15 taxa with a significant intergroup variation.

Taxa Name	Group with Higher Abundance	LDA_Score	*p*-Value
Phylum			
*Firmicutes*	HS	4.707	0.018
Class			
*Alphaproteobacteria*	CN	4.178	0.018
*Negativicutes*	HS	3.768	0.0484
Order			
*Micrococcales*	CN	4.365	0.048
*Rhizobiales*	CN	3.772	0.035
Family			
*Ruminococcaceae*	CN	4.429	0.002
*Lachnospiraceae*	CN	4.209	0.0023
*Microbacteriaceae*	CN	3.896	0.004
*Clostridiaceae_1*	HS	4.549	0.035
*Veillonellaceae*	HS	3.769	0.025
Genus			
*Subdoligranulum*	CN	4.314	0.002
*Lactococcus*	CN	4.107	0.001
*Ruminococcus__torques_group*	CN	3.913	0.002
*Leucobacter*	CN	3.854	0.004
*Proteiniclasticum*	HS	4.263	0.002

Larger LDA scores indicate more significant differences.

## Data Availability

Not applicable.

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
