# Peer review of "Chronic Heat Stress Affects Bile Acid Profile and Gut Microbiota in Broilers"

_ijms, 2023, doi:10.3390/ijms241210238_

Round 1
Reviewer 1 Report
The results of the present investigation showed that chronic HS alters the gut microbiota of broilers and disturbs the equilibrium of bile acid metabolism. The study is intriguing and can be approved when the following issues have been addressed.
There are many abbreviations in the abstract need to be write in the full name in the first use and then abbreviated.
Line 36-46 require citations
Discussions usually get too broad. It does not provide a compelling biological justification for the large differences you observed such as chronic HS alters the gut microbiota of broilers and disturbs the equilibrium of bile acid metabolism in your study. Updated citations up to 2023 and discussions of pertinent previously published material were requested in order to comprehend the significance of the study's findings.
chronic HS disrupts the homeostasis of bile acid metabolism in broilers, which is associated with alterations in gut microbiota.
Author Response
Dear Reviewer,
Thanks for all the comments. Please find the attachment for the detail reply.

Reviewer 2 Report
The authors reported that chronic HS disrupts the homeostasis of BA metabolism in broilers, which is associated with alterations in gut microbiota. It is a well-designed study. The following issue should be improved.
(1) In the abstract, D56 should be specifically into the 56-day age.
(2) Were the broilers free to water and feed? The data on the feed intake and water consumption should be provided.
(3) A diagram for the experimental design is suggested to provide.
(4) The magnification and ruler bars for Figures 1c, and 2a should be provided.
None
Author Response
Dear reviewer,
Thanks for all the suggestions. Please find the attachment for the detail reply.

Reviewer 3 Report
Comments to the Author
In general
This is a study of using broiler chickens as model to discern if chronic heat stress affects bile acid profile and gut microbiota. The study raise some interesting questions regarding the detrimental effects of heat stress on bile acids and extent of and antioxidant activity and lipid oxidation, and how increasing the temperature can affect the bile acids synthesis under heat stress conditions. The manuscript is written well. Specific comments are mentioned below.
Introduction
The cited results of the previous experiments are somewhat old. I suggest pointing to newer data, as this is a very up-to-date topic.
Line 31: This is the first time to mention BA, please mention the full name
Line 45: Please do not begin the sentence with abbreviation.
Materials and methods
I think this section should be mentioned after the introduction directly.
My great concern here about using 36 ± 1 ℃ to induce heat stress is too much, in think 32 may be enough, could you please explain why?
Discussion
There is no information on the basic mechanisms related to the bird health under thermal stress. Furthermore, this study concerns of meat of the birds, I suggest briefly describe the impact of thermal stress on the meat quality.
Author Response

(The authors gave the same response as above.)
